# The Impact of Domain Shift on Predicting Perceived Sleep Quality from Wearables

**DOI:** 10.3390/s25134012

**Published:** 2025-06-27

**Authors:** Nouran Abdalazim, Leonardo Alchieri, Lidia Alecci, Pietro Barbiero, Silvia Santini

**Affiliations:** Faculty of Informatics, Università della Svizzera Italiana, 6900 Lugano, Switzerland; leonardo.alchieri@usi.ch (L.A.); lidia.alecci@usi.ch (L.A.); pietro.barbiero@usi.ch (P.B.); silvia.santini@usi.ch (S.S.)

**Keywords:** personal informatics systems, sleep behaviour monitoring, sleep quality recognition, dataset bias, domain shift, covariate shift, online learning, wearable devices, machine learning models

## Abstract

Machine learning models for personal informatics systems are typically trained offline on *records of a specific population of users*, resulting in *population models.* These models may suffer performance degradation in real-world settings due to *domain shift*, i.e., differences in data distributions across users and contexts. Domain adaptation techniques can address this *issue* by, *e.g.,* personalizing models with user-specific data. *In this paper, we quantify the impact of domain shift* on *the performance* of both population and personalized models *in a specific scenario:* sleep quality recognition. *To this end, we also collect and make available to the research community the new BiheartS dataset*. Our analysis shows *that* domain shift *causes the* accuracy of population models *to decrease* by up to 18.54 percentage points, when *used* on *new data*. Personalized models, *instead*, show robust performance across datasets. However, *crafting personalized models typically requires using new data or user-provided labels*, limiting their *applicability in real settings*. To *mitigate* the limitations *of both population and personalized models*, we propose a novel unsupervised domain adaptation approach: the cluster-based population model (CBPM). CBPM achieves accuracy improvements of up to 13.45 percentage points *w.r.t. population model* without requiring *the use of* user-specific records or *labels*.

## 1. Introduction

Personal informatics (PI) systems are ubiquitous computing technologies designed to help users collect, analyze, and reflect on personal information to enhance self-awareness and achieve personal goals [1]. These systems support self-reflection and self-knowledge by providing the users with insights about their daily behaviour and practices, e.g., health, productivity, emotions, and well-being [2]. PI systems often rely on the use of wearable devices, e.g., smartwatches, digital rings, earbuds, to collect raw data about their users’ actions, physiological responses, emotional states, context, and more.

Machine learning (ML) models are often used in PI systems to infer high-level constructs from sensor data, for instance mental health [3,4,5,6], sleep behaviour [7], students’ emotional engagement in classrooms [8,9], or behavior in the workplace [10,11,12]. To train ML models used in PI systems, researchers and practitioners typically rely on existing datasets containing the data of a population of users. The obtained model is referred to as a *population model* or *one-size-fits-all model* [4] and can be applied unchanged to perform inference from the data collected about the (previously unseen) target user of a PI system.

However, the distribution of both input data and labels may shift significantly across users within the population used to train the model and the target unseen user(s). This issue, known as *domain shift* in the ML literature, is caused by *interpersonal variability* across users and may cause population models to perform poorly in real settings [4,13]. To cope with this problem, *domain adaptation* techniques can be used to bridge the distribution mismatch between the training data and the data of the target user(s). For example, a pre-trained model can be adapted to the specific characteristics or data of the target user, leading to what in the PI systems literature is typically referred to as a *personalized model* [7,14].

In this work, we provide a comparative analysis of the performance of personalized models in a specific use case of PI systems: sleep monitoring from human physiological signals. Specifically, we first assess the impact of personalization on model performance using data records from a single dataset, as has been carried out in similar works in the literature [4,7,15,16]. We thereby use a subset of the data for model training and the remaining data for model personalization and evaluation. Yet, since the records pertain to the same dataset, they tend to have lesser interpersonal variability than if the data was picked from different datasets—a phenomenon also known as *dataset bias* [3,17,18,19].

Thus, we repeat our analysis using entirely different datasets for model training, personalization and evaluation. To this end, we rely on both the existing M2sleep dataset by Gashi et al. [7], and on the novel BiheartS dataset which we collected in the context of this work and make publicly available to the research community. BiheartS contains data records of 10 participants collected in a real-world setting over 30 days. The records include physiological signals gathered using three wearable devices—an Oura ring (third-generation) and two Empatica E4 wristbands (worn one on each wrist)—as well as self-reports about participants’ sleep and daily behaviour.

Lastly, we describe a novel domain adaptation technique that does not require labelled data from the target user for personalization. The method matches each sleep session of the target user to one of previously determined *clusters* of training data, thereby capturing both interpersonal variability between users as well as intra-personal variability of the target user. Since the method generates a cluster-based population model, we refer to it as CBPM.

Our results indicate that personalized models outperform population models by up to 6.81 percentage points. When different datasets are used for evaluation, the performance of personalized models attains similar levels with respect to the single-dataset scenario. However, the performance of population models drops by up to 18.54 percentage points. This highlights the risk of overestimating the performance of population models when dataset bias is not taken into account. If our CBPM technique is used instead, the mean accuracy of the population model increases by up to 13.45 percentage points even when different datasets are used for training and evaluation.

The remainder of this paper is organized as follows: Section 3 summarizes relevant background information and related work. Section 3 describes the BiheartS dataset, contributed as part of this study, whereas Section 4 describes the M2Sleep dataset and the data pre-processing procedures. Section 5 provides evidence of the existence of dataset bias between the BiheartS and M2sleep datasets, proving that they are adequate for our analysis. In Section 6, we show how dataset bias causes significant performance degradation of population models, whereas personalized models can adapt effectively to the data of target users irrespective of the setting. Lastly, in Section 7, we introduce our CBPM technique and analyse its performance. Section 8 discusses the obtained results and their implications, whereas Section 10 concludes the paper.

The BiheartS dataset is available upon signing a data sharing agreement. All software artifacts developed as part of this work are open-source and available at the following GitHub (vesion 1) repository: https://github.com/nouran-abdalazim/sleep_quality_recognition_models.git (accessed on 17 June 2025). The link contains an excerpt of the repository for reviewing purposes. The full code base is going to be released upon acceptance of this paper).

## 2. Background and Related Work

Since our research centers on domain adaptation for sleep quality recognition, we examine the pertinent literature on domain adaptation broadly, as well as specifically for sleep recognition.

### 2.1. Domain Adaptation

In the ML literature [17,20,21,22], a *domain D* consists of the input feature vectors *X*, the output labels *Y*, and the joint probability P(X,Y), i.e., D=(X,Y,P(X,Y)). The domain over which a model is trained is referred to as the *source domain*
$D_{s}=\left(X_{s},Y_{s},P_{s}\left(X_{s},Y_{s}\right)\right)$, whereas the domain over which the model is tested (or used in real settings) is instead referred to as the *target domain*
$D_{t}=\left(X_{t},Y_{t},P_{t}\left(X_{t},Y_{t}\right)\right)$. In the PI systems literature, such a model is typically referred to as a *population model* [4] because it captures the characteristics of the population of users over which it has been trained.

The differences between the distribution $P_{s}\left(X_{s},Y_{s}\right)$ in the source domain, which is used for model training, and the distribution $P(X_{t},Y_{t})$ in the target domain, where the model is deployed, leads to the so-called *domain shift* problem [13,23]. This mismatch between the source and target distribution makes the ML model unable to perform reliably when confronted with the new data in the target domain, which was unseen at the time of training. In other terms, the model does not *generalize* well to the target domain(s). There are different types of domain shift: covariate shift, label shift, and concept shift [13].

Our work focuses in particular on *covariate shift*, which occurs when the distribution of the marginal probability of the feature vectors in the source domain $P(X_{s})$ differs from the same distribution in the target domain $P(X_{t})$. In PI systems, this occurs for instance when the distribution of the features computed from physiological data in the source domain differ systematically from the distribution of features in the target domain due to, e.g., significant age differences in the population of users whose data was included in the source domain and the age of the target user. In general, variability in users’ demographic characteristics, behaviour, preferences, contexts, and perceptions can lead to covariate shift.

Domain adaptation (DA) is a paradigm that has been widely used in the computer vision and natural language processing literature to deal with the covariate shift problem [24,25,26,27]. DA techniques typically exploit the feature vectors and/or labels from the target domain to adapt the model that is previously trained on the source domain. In the PI systems literature, the model resulting from this adaption is usually referred to as a *personalized model* [4,7].

There exist several categories of DA techniques depending on the specific differences between the feature vectors and/or the corresponding labels in the source and target domains.

Wang and Deng [25], for instance, specify that DA can be applied either in *homogeneous* settings, where the input feature space and the label space are the same in the source and target domains, or in *heterogeneous* settings, where the feature space and the label space differ across domains. Also, Farahani et al. [24] refer to the setting where the label space and the feature space in the source and target domains coincide as a *closed-set* setting [24].

Further, Patel et al. [26] highlight that DA techniques can be categorized based on the availability of labelled feature vectors from the source and the target domains. In particular, they differentiate between semi-supervised and unsupervised approaches. In semi-supervised DA approaches, the ML model exploits labelled data from the target domain to adapt itself. In PI systems, this implies that the target user(s) can provide labels for the data collected by the system after the deployment of the ML model. In unsupervised DA approaches, the model leverages only unlabelled data from the target domain. In PI systems, this implies that new user data collected by the system after the deployment of the ML model is used for adaptation, but that the corresponding labels are not available.

Lastly, Wang and Deng [25] also show that DA can occur either in one step or in multiple steps, depending on whether transferring knowledge from the source to the target domain can be accomplished in one or multiple steps.

In this paper, we address the covariate shift problem using DA in homogeneous closed-set settings. We explore the use of both a personalized model generated using a one-step, semi-supervised approach and a cluster-based model derived by using a multi-step, unsupervised DA technique.

### 2.2. Domain Adaption in Sleep Quality Recognition

In the PI systems literature in general, and in sleep quality recognition in particular, several authors have explored DA techniques to cope with the poor generalizability of population models [7,15,16,28].

Gashi et al. [7] collect and use physiological signals along with self-reports to build and evaluate ML models for sleep quality recognition and sleep/wake recognition tasks. They compare the results of two models—population models and personalized models—using the M2sleep dataset. Their experimental results confirm that personalized models achieve higher performance than population models in both tasks. In particular, the personalized model achieves a maximum balanced accuracy of 61.51, with an improvement of 14 percentage points in the sleep quality recognition task. Also, Moebus and Holz [28] use the M2sleep dataset proposed by Gashi et al. [7]. They add features that represent cardiovascular activity instead of the features extracted from electrodermal activity as Gashi et al. [7]. Their results reveal the personalized model achieves a maximum accuracy of 70%, with an improvement of 11 percentage points in the model performance compared to Gashi et al. [7]’s approach. Van et al. [16] use multiple datasets to estimate sleep efficiency. They rely on activities, emotions, environmental, and sleep data collected using the Fitbit wearable device in their ML task. They train an ML model on one dataset (population model), and then they adapt this model based on the data of the target test user (personalized model). They derive the final estimation from the results of both population model and personalized model. Their results show that the integration of the population model and the personalized model achieves an improvement of 0.5 in root mean square error compared to the baselines. Khademi et al. [15] use actigraphy data to build a personalized model to distinguish sleeping/waking events and estimate sleep parameters from golden standard polysomnography (PSG). They compare a personalized model, built using the data of the target user, with a population model. Their experimental results show that the performance of the personalized model is higher than that of the population model in estimating sleep parameters. The results show no statistical difference between the sleep parameters estimated by the personalized model and the PSG. Gan et al. [29] propose a framework that integrates objective and subjective data from the Fitbit wristband, a sports logging app, and questionnaires, incorporating daily activity, affective states, sleep information, and demographic data. Specifically, they train a deep learning model using objective and subjective data to estimate sleep quality. Moreover, they use pattern mining to discover different relations between the data from different modalities and sleep quality. Then, they rely on the model output and pattern mining to provide the personalized feedback about sleep habits to the users.

Grandner et al. [30] compare two ML models adopted on a new wearable device. The two models are a population model and a personalized model in sleep stage recognition and sleeping/waking recognition tasks. They use the parameters from the golden standard PSG as the ground truth. Their results show that the personalized model surpasses the population model.

### 2.3. Novelty of This Work with Respect to the Existing Literature

The use of DA techniques to cope with the domain shift problem has been widely explored in the computer vision and natural language processing literature [17,20,21,22]. However, less attention has been paid to this problem in PI systems [3,13]. In particular, in sleep quality recognition, several studies in the literature attempt to investigate the impact of DA i.e., personalized model, e.g., [7,15,16,28]. However, the majority of the existing studies employ only one single dataset in their experimental setup. Such an experimental setup is less prone to the effect of domain shift, since users in the same dataset tend to share common characteristics. This can lead to misleading over-estimated results for the performance of the personalized model.

In this study, we evaluate the performance of both the personalized and population models in a multiple-dataset experimental setup. To this end, we introduce the new BiheartS dataset, collected using a data collection protocol similar to that of the existing M2Sleep dataset. These two datasets enable us to investigate the presence of covariate shift between datasets with comparable characteristics, e.g., wearable devices, geographical regions, and demographic characteristics of participants. Furthermore, we propose a novel cluster-based approach that leverages unsupervised DA to enhance the performance of the population model across diverse users in different datasets, eliminating the need for user feedback in order to adapt the model.

## 3. BiheartS Dataset

To analyze the impact of distribution mismatch across datasets, we collect a new dataset named BiHeartS. With the BiheartS dataset, we provide the research community with a new dataset that includes extensive self-reports about the daily behaviour, physical activity, daily fatigue, daily stress and the sleep behaviour of participants. The design of the data collection protocol aligns with existing studies in the literature [7,31,32]. However, some key novel elements in the BiheartS dataset are the use of two wearable devices, which collect sensors’ data from both sides of the body simultaneously, and the extensive daily self-reports. To the best of our knowledge, the BiheartS dataset is the first dataset that is collected in the wild and includes sensors’ data from both sides of the body. The BiheartS dataset is available after signing a data sharing agreement. Please contact the corresponding author of the paper to make a request.

Given the extensive self-reports and sensor signals collected in the BiheartS dataset, researchers can use this dataset in several research studies. They can use it to investigate personalized sleep quality recognition, similar to [7,28]. The BiheartS dataset also includes raw sensor data from multiple wearable devices, enabling direct comparisons between different wearables, similar to [31,33]. Additionally, since BiheartS contains raw sensor data from both the left and right sides of the body, researchers can explore lateralization analysis, as in [34,35]. In this paper, we focus on the sleep quality recognition task. The data collection campaign has been reviewed and approved by our faculty’s delegate for ethics.

**Participants and Devices** We recruit 10 participants (1 female and 9 males) aged between 20 and 25 years old. All the participants are students. We use the term user and participant interchangeably in this work. Participants are asked to wear one finger-worn wearable device and two wrist-worn wearable devices for 30 consecutive nights: the Oura ring (Gen. 3) (https://ouraring.com/product/rings), (accessed on 17 June 2025) which measures sleep parameters with a Photoplethysmography (PPG) sensor, and two Empatica E4 (https://www.empatica.com/en-gb/research/e4/) (accessed on 17 June 2025) wristbands, one on each wrist. The Empatica E4 is a research-grade device equipped with four sensors: electrodermal activity (EDA), an accelerometer (ACC), skin temperature (ST), and PPG sensors [36]. The E4 wristband and the Oura ring, especially the second generation, are adopted in several studies in the literature, e.g., [7,31,33,37,38,39,40,41].

**Self-reports** To collect the information about the participants, their sleep behaviour and their daily routine, we rely on self-reports, similar to existing studies in the literature, e.g.,  in [7,31,32]. We divide the self-reports into a *pre-study questionnaire*, *daily self-reports*, and a *post-study questionnaire*. The pre-study questionnaire integrates demographics questions and several validated questionnaires: the Pittsburgh Sleep Quality Index (PSQI) [42], to assess sleep behaviour; the Morningness–Eveningness (Chronotype) questionnaire, to determine the chronotype [43]; and the Big 5 inventory questionnaire to estimate personality traits [44].

We provide the participants with two options to log their daily self-reports: a pen-and-paper diary and an application-based diary, the RealLife Exp mobile application (https://www.lifedatacorp.com) (accessed on 17 June 2025). We split the daily self-reports into *morning self-reports* and *evening self-reports*. The morning self-reports include information about the sleep behaviour of the previous night: bedtime, wake-up time, latency, number of awakenings, perceived sleep quality score, recovery after sleep score, and reasons for sleep disturbance. All the questions in the morning self-reports are formulated based on the validated PSQI questionnaire [42] and the Daily User Caregiver Sleep Survey (DUCSS) questionnaire [45]. We perform adaptations to some of the questions from the validated questionnaire to match our study design, e.g.,  we adapt the question *“During the past month, How long (in minutes) does it usually take you to fall asleep each night?”* from the PSQI questionnaire to be *“How long (in minutes) did it take you to fall asleep last night?”*. We present more details about the questions used in the morning self-reports and the corresponding adaptions in the Appendix A. The evening self-reports include information about the routine and the behaviour during the current day: stress score, sleepiness score, fatigue score, physical health, and physical activity. For the evening self-reports, we adapt questions from the following questionnaires: the Perceived Stress Scale Questionnaire [46], PSQI questionnaire [42], the Quality of Life Enjoyment and Satisfaction Questionnaire [47], and the International Physical Activity Questionnaire [48]. Finally, the post-study questionnaire includes the PSQI questionnaire to assess the overall sleep behaviour of the participants over the whole study duration. We designed the data collection campaign and the questions in the self-reports in collaboration with an expert in psychology and organizational behaviour. We illustrate the details of the morning and the evening self-reports used in the BiheartS data collection campaign in the Appendix A.

### Data Collection Procedure

We design the data collection procedure following similar studies in the literature, e.g., [7,31,32,33]. In the beginning, we provide the participants with information about the objective of the study and the study protocol, then the participants sign an informed consent form. Finally, we give the wearable devices to the participants alongside instructions for setting up the designated synchronization applications for obtaining the raw data from the wearable devices.

Before the study, the participants respond to a pre-study questionnaire. Furthermore, we request that the participants put the pen-paper diary beside their beds so they can easily remember to log their daily self-reports. We use the RealLife Exp mobile application to send two notifications in the morning and two notifications in the evening to the participants as a reminder to compile the daily self-reports. To minimize the effect of the recall bias, the participants are allowed to log only self-reports of the current day.

We ask the participants to wear the Oura ring on the index finger of their left hand. They wear one Empatica E4 on their left wrist and one on the right wrist, every night. We call the sleep period of a participant during the night a *sleep session*. The participants wear the devices one hour before sleep and log the evening self-reports. The next day, participants complete the morning self-report and then take off the devices one hour after waking up. During the day, participants synchronize the data of each device. To ensure that the participants synchronize the data from the wearable devices on a daily basis, we run daily checks. After the study, the participants respond to the post-study questionnaire.

## 4. Datasets and Data Preparation

To investigate the effect of domain shift in PI systems, we use two datasets: the BiheartS dataset, which we collect for the purpose of this study, and the M2sleep dataset [7]. Both the BiheartS and the M2sleep datasets are collected for sleep behaviour analysis in the wild with similar data collection protocol. They are collected using similar wearable devices, in the same geographical zone, from participants with a comparable age range and occupation. The similarities between the two datasets make them suitable for studying covariate shift in the same task: sleep quality recognition. Table 1 shows the summary of the two datasets.

We describe the BiHeartS dataset in the dedicated Section 3. Moreover, Figure 1 presents a 30-second sample of raw signals collected from one participant in one sleep session from the BiHeartS dataset. Moreover, we briefly present the M2sleep dataset in this section, as well as the pre-precessing procedure for both datasets.

### 4.1. The M2sleep Dataset

The M2sleep dataset, collected by Gashi et al. [7], is widely used in studies involving sleep behaviour in the literature, e.g., [7,28,49]. It is publicly available upon signing a data sharing agreement with the authors. It contains data collected from 16 participants for 30 nights in the wild. Some 69% of the participants are male, and 56% of the participants are students in the age range of 21–35 years. The dataset includes sensor data collected using the Empatica E4 wristband, along with self-reports about the previous sleep session including bedtime, wake-up time, awakening, and perceived sleep quality.

### 4.2. Pre-Processing, Feature Extraction, and Labelling

In our experiments, we rely on sensor signals collected from the Empatica E4 wristband: heart rate (HR), inter-beat interval (IBI), ST, and ACC signals. We also extract the respiration patterns (RP) from the photoplethysmogram (PPG) signal using a similar methodology as described in [50]. We refer to each of the IBI, ST, ACC and RB signals as *modality*). These modalities are widely used in sleep quality recognition studies, e.g., [7,28,31,51,52]. From each sleep session—i.e., the sleep period from one participant—we extract hand-crafted features from each modality individually. We refer to the set of features extracted from the same sensor signal as a *single modality*. For HR, ACC, ST and RP signals, we extract eight time-domain features for the whole signal as in [7] and four frequency-domain features by transforming each signal using the fast Fourier transform (FFT), similar to [7]. We use the Scikit-Learn [53] Python library to extract these features. To extract features related to heart rate variability (HRV) from the IBI signal, we “clean” the signal by removing IBI values outside the 300–2000 ms range [54]. Then, we remove the ectopic beats based on Malik’s rule, as suggested in [54]. Finally, we extract 12 time-domain features and 7 frequency-domain features using the FLIRT [54] and the hrv-analysis (https://aura-healthcare.github.io/hrv-analysis/readme.html accessed on 12 June 2025) Python libraries.

In addition to features extracted from single sensor modalities, we employ contextual features from self-reports. These features are objective measures for each participant’s previous sleep session. Several studies in the literature show that contextual features are correlated with perceived sleep quality [14,28]. Consequently, they can be used as an indicator of the participants’ perception of their sleep quality.

Moreover, we group features from the single sensor modalities to form sets of features, which we refer to as *multi-modality*. The first multi-modality set consists of features from all sensor modalities together, which we define for simplicity as the *all-sensor-features modality*. The second set combines features from all the sensor modalities as well as the contextual modality, which we refer to as *all-features modality*. Several studies in the literature rely on the multi-modality in their experiments on sleep quality recognition [7,28]. In Table 2, we present a summary of all the modalities used in our experiments. We provide more details for the feature extraction from each modality in the Appendix A.

We formulate the sleep quality recognition task as a binary ML task, similar to Gashi et al. [7], Moebus and Holz [28], Abdalazim et al. [31], Moebus et al. [49]. Participants in both datasets report their sleep quality score using Likert scales; for the M2Sleep dataset, the scale is from 1 (lowest quality) to 5 (highest quality), while for the BiHeartS, the scale is from 1 (lowest) to 10 (highest). In order to obtain binary labels, following similar works in the literature [7,28,31], we consider the following. For the M2Sleep dataset, we select as “high sleep quality” all scores of 4 or above (top 20% of the 5-point Likert scale), and as “low sleep quality” all scores of 3 or less, similar to [7,31]. For the BiHeartS dataset, scores of 7 or above (top 20% of the 10-point Likert scale) are labelled as “high sleep quality”, while those of 6 or less are “low sleep quality”.

By applying the same feature extraction procedure to both datasets and the same threshold to define positive and negative labels, we ensure the homogeneity of the feature and the label spaces. This allows the investigation of the covariate shift and the cross-dataset generalization capability of ML models, as recommended by Xu et al. [3], Meegahapola et al. [13].

## 5. Domain Shift Analysis

In this section, we describe how we assess the domain shift between the two datasets, BiHeartS and the M2sleep. To this end, we first conduct a statistical analysis across the datasets, as is common practice in the literature, e.g., [3,19], using correlation and statistical tests. Then, we perform a name-the-dataset ML classification task to estimate the degree of dataset shift between the feature vectors in the M2sleep and the BiheartS datasets [3,17].

### 5.1. Statistical Comparison Between Datasets

First, we compute the Spearman’s rank correlation coefficient between the perceived sleep quality as reported on the Likert scale in the morning self-reports and each extracted feature. With this experiment, we aim to investigate the variability of the correlation patterns across the features from both datasets. This variability is an indication of the difference in the features’ distribution across the M2sleep and the BiheartS dataset. We present in Figure 2 a scatter plot that shows the relation between the correlation coefficient of each feature in the two datasets. We observe that the correlation coefficient of the majority of the features is different in both datasets. These results hint at differences in the distribution of the features across the datasets, and, accordingly, these differences can lead to covariate shift. Consequently, we compare the distribution of each feature statistically in the two datasets. We provide more results for the correlation analysis for each feature in the Appendix A.

We perform the two-sample Kolmogorov–Smirnov non-parametric statistical test (α=0.05) [55] between the distributions of each feature when extracted from the BiHeartS and the M2Sleep datasets. Figure 3 presents the distribution of the p-values computed for each feature, divided by modality, between the two datasets when using the Kolmogorov–Smirnov statistical test. These results show that for all the modalities, except the ACC modality, the majority of the features are statistically different in the two datasets. In particular, we observe that 61.11% of the features are statistically different between the two datasets. Also, we perform Bonferroni corrections to the α, such that $\alpha_{n}=\frac{\alpha }{n}$, where *n* is the number of features. When performing the correction, we find that 27.78% of the features are statistically different, with $\alpha_{n}\approx 0.00069$. These results confirm the existence of a distribution mismatch between most of the features and imply the existence of covariate shift between the two datasets.

### 5.2. Name-the-Dataset Machine Learning Task

In the computer vision literature, the presence of domain shift is often evaluated using the name-the-dataset (NED) ML task [17,18]. Xu et al. [3] used the same task to quantify the difference among datasets in the human behaviour modelling domain. Accordingly, we employ the NED task on our two datasets, in order to evaluate the ability of ML models to distinguish between the feature vectors of the two datasets. We formulate the task as a binary classification task, and we use features from the *all-features modality* described in Section 4.2. We use a support vector machine (SVM) classifier, as suggested in [17,18]. We also sample varying proportions of data from each dataset, similar to [17,18], while ensuring that users present in the training set are not in the test set, as in [3]. As an evaluation metric, we use the balanced accuracy.

Figure 4 shows the performance of the SVM classifier and the RG baseline. We conduct this experiment multiple times with a different number of random samples in each round to investigate the behaviour of the SVM classifier, as in [18]. With this different number of samples, we can observe the change in the performance of the SVM classifier as more data from each dataset is added to the training set. From Figure 4, we find that the balanced accuracy of the SVM classifier increases as more samples from both datasets are included in the experiment. Moreover, the difference in the balanced accuracy between the SVM and the baseline is statistically significant when measured using the Wilcoxon statistical test (α=0.05).

The results of the NED classification task support our observation about the distribution mismatch between the two datasets, since the SVM classifier can distinguish between the feature vectors from the two datasets with a balanced accuracy of 90.72% when all the samples from the two datasets are used. This implies the existence of covariate shift between the two datasets. Accordingly, we explore the impact of covariate shift on the performance of population models and personalized models in a sleep quality recognition task.

## 6. Impact of Domain Shift on Sleep Quality Recognition

In the previous Section 5, our results demonstrate the existence of covariate shift between the BiheartS and the M2sleep datasets. In this section, we investigate the impact of covariate shift on the performance of ML models in a sleep quality recognition task. To this end, we rely on different evaluation scenarios, two training settings, and two different ML classifiers. We rely on the feature sets from multiple modalities —all-sensor-features and all-features— as well as the contextual modality, given that existing studies demonstrate that they achieve higher results than single-modality feature sets [7,28]. Also, the multi-modality feature set allows us to assess whether our findings are consistent across different types of input features.

### 6.1. Evaluation Scenario

We implement three distinct setups (hereafter *evaluation scenarios*) to train and evaluate the sleep quality recognition models. The three evaluation scenarios are *single-dataset*, *mixed-dataset*, and *multiple-dataset*. With these scenarios, we have three different setups for the datasets used for model training and evaluation.

***Single-dataset scenario***: This evaluation scenario relies on the same dataset for model training and evaluation. The implementation of this scenario is similar to the leave-one-participant-out evaluation method, where in each training/evaluation iteration, one user from the designated dataset is left out as the target test user for model evaluation.

***Multiple-dataset scenario***: It is implemented as the leave-one-dataset-out evaluation method, where one dataset is left out at each training/evaluation iteration. The model is then evaluated on each user in the left-out dataset. In other words, this scenario is a composite of the leave-one-participant-out evaluation method and the leave-one-dataset-out evaluation method.

***Mixed-dataset scenario***. In this evaluation scenario, the two datasets are combined together as one dataset. Then, the leave-one-participant-out evaluation method is used to evaluate the ML model per user, i.e., the left-out target user. This scenario acts as a middle ground between the single-dataset scenario and the multiple-dataset scenario, as it ensures that the training set of the model includes feature vectors from the same dataset as the target left-out user. In this scenario, the model is trained on the data from users within the same dataset as the target user as well as the data from all the other datasets.

In all the previous scenarios, the data from each user is sorted in chronological order to avoid any temporal data leakage. We provide more details and mathematical formulations for the three scenarios in the Appendix A.

### 6.2. Training Settings and Baselines

To investigate the impact of covariate shift on sleep quality recognition models, we implement two possible real-world settings for PI systems. The first setting relies on ***population models*** that are trained once using the available training data then deployed without adjustments to the target unseen users. In our experiments, the population model is trained on the available training set depending on the scenario (Section 6.1). Then, it is evaluated on target unseen users in the test set. This model is an offline model because the model is used by the target users without any adjustments.

The second setting relies on the *online learning* approach [56] to adapt the model over time for the target test user. This setting allows the users to provide their feedback about the insights provided by the PI systems. Then, this feedback is used to adjust the model, which we call, in this case, *personalized models*. In our experiments, the personalized model is trained using the available training set based on the scenario. Then, the model is updated incrementally for the target unseen user when a new labelled data point from this designated user is available. This model relies on the discrepancy-based domain adaptation approach with a class criterion [25]. The personalized model is considered online model because the model’s weights are updated with the arrival of a new data point.

To implement both models, we use two different ML classifiers in our experiments. The first one is the Passive Aggressive Classifier (PAC) (https://scikit-learn.org/1.5/modules/generated/sklearn.linear_model.PassiveAggressiveClassifier.html) (accessed on 17 June 2025) [57]. The PAC is widely used in online learning settings [57], in particular for personalization purpose as demonstrated by existing studies, e.g.,  Winger et al. [58,59]. The second one is the Multilayer Perceptron (MLP) (https://scikit-learn.org/1.5/modules/generated/sklearn.neural_network.MLPClassifier.html#sklearn.neural_network.MLPClassifier) (accessed on 17 June 2025) [60,61], also known as a feed-forward neural network and one of the simplest deep learning techniques [62]. The MLP classifier is employed in several studies focused on monitoring sleep behavior and classifying sleep quality using data from wearables [62,63,64]. For instance, Sathyanarayana et al. [62] report that the MLP achieves performance comparable to deep learning models such as Convolutional Neural Networks (CNNs) and Long Short-Term Memory (LSTM) networks in recognizing sleep quality from actigraphy signals. This finding is further supported by [64]. Based on this evidence, we include the MLP classifier in our experiments, particularly given our context, which does not involve large datasets that typically benefit complex deep learning models. We implement both models using the Scikit-Learn [53] Python  library (version 1.6.1).

We employ three baselines: the random guess (RG) baseline, the biased random guess (BRG) baseline, and the personalized biased random guess (PBRG) baseline. The *random guess baseline* makes sleep quality predictions by randomly extracting positive and negative labels from a uniform distribution. The *biased random guess baseline* makes predictions by randomly selecting the positive and negative labels from the labels’ distribution in the training set. Both the RG and the BRG baselines are offline models, since the distribution of the labels in the training set does not change. Finally, the *personalized biased random guess baseline* is a variation of the BRG baseline. It extracts the predictions by considering both the distribution of the labels in the training set and the new available data point from the target test user. This baseline is an online model, since it considers data from the target user to update the label distribution in the training set.

### 6.3. Evaluation Metrics

Offline models are typically evaluated using the leave-out method, which splits the dataset into training and test sets [56]. Online models, however, are often assessed with the predictive sequential (prequential) method [56,65,66], also known as interleaved test-then-train. This approach evaluates the model on each test data point before using it for training, enabling performance assessment over time.

However, to train and evaluate the performance of online models and offline models simultaneously, Fekri et al. [56] propose the prequential–leave-out evaluation method. This method uses the available training data for training the offline models and the online models. However for model evaluation, the online models are evaluated using the prequential evaluation method and the test data, while the offline models are evaluated using the traditional hold-out evaluation method using the same test data. This approach guarantees that regardless of the available training data for both models, the two models are evaluated using the same test data.

In our experimental setup, we rely on the prequential–leave-out evaluation method to evaluate the offline models—the population model, biased random guess, and random guess—and the online models, i.e., the personalized model and personalized biased random guess.

For evaluation, we use accuracy as the metric for both online and offline models. Prequential accuracy is commonly used to assess online models [66]; however, it is mathematically equivalent to accuracy when computed on the entire test set, assuming that the target test user provides feedback for each inference result.

Moreover, to statistically compare the performance of the population model and the personalized model in the different scenarios, we use Cliff’s δ effect size [67]. Cliff’s δ is non-parametric effect size measure, with a range of [−1, +1], that quantifies the overlap between two distributions, where values near the two extremes indicate minimal overlap and values close to 0 indicate high overlap. We use the thresholds proposed by Vargha and Delaney [68] to interpret the effect size values as negligible, small, medium, and large.

### 6.4. Experimental Results for Sleep Quality Recognition

We aim to recognize the binary sleep quality labels in three evaluation scenarios (a single-dataset scenario, a multiple-dataset scenario, and a mixed-dataset scenario) and two model training settings (a personalized model and a population model). In each scenario, we evaluate the population model and the personalized model using seven different modalities. To mitigate the effect of random initialization of the models, we run each experiment using 50 different random seeds, and we compute the average accuracy over all iterations for each model per each modality. In this section, we present the results for the PAC and the MLP classifiers in the single-dataset scenario and the multiple-dataset scenario with the all-sensor-features, the contextual, and the all-features modalities in Table 3. Moreover, we report the difference in the models’ performance across the different scenarios in Table 3. We present more results on the mixed-dataset scenario and the other modalities in Appendix A.

***Sleep Quality Recognition in the Single-Dataset Evaluation Scenario.*** The personalized models generally achieve higher performance than the population models across most modalities in both datasets. Additionally, both models outperform the baseline performance in most cases. Specifically, for the M2Sleep dataset, the personalized model achieves the highest accuracy of 64.95% using the all-features modality with the MLP classifier. In the BiheartS dataset, it reaches a maximum accuracy of 70.39% with the same modality and the PAC classifier. On the other hand, the population model attains its best performance on the M2sleep dataset with 60.69% accuracy using the contextual modality and MLP classifier, and 64.41% accuracy on the BiheartS dataset using the all-sensor-features modality and PAC classifier. These results show that personalized models outperform population models in most cases. However in some cases, both models have comparable performance—for instance, using the all-sensors-features modality and the MLP classifier for the BiheartS dataset and using the contextual modality and the MLP classifier for the M2sleep dataset. This implies that when the target test user belongs to the same dataset used for model training, both the personalized model and the population model can achieve comparable performance. Accordingly, the impact of the covariate shift is not evident in the single-dataset scenario.

***Sleep Quality Recognition in the Multiple-Dataset Evaluation Scenario.*** We find that the personalized models consistently outperform the population model across all modalities in both datasets, achieving the highest accuracies of 64.12% and 65.66% using the contextual modality with the PAC classifier for the M2sleep and the BiheartS datasets, respectively. Moreover, they outperform the RG and BRG baselines in most cases. On the other hand, for the population model, there is a drop in the performance of the population model in this scenario compared to the single-dataset scenario. For instance, the population model achieves a maximum accuracy of 57.40% with the M2sleep dataset using the MLP classifier and the all-sensor-features modality. Also for the same modality and the BiheartS dataset, it reaches a maximum performance of 49.63% using the PAC classifier.

These results imply that the leading performance of the personalized model is consistent across the different scenarios. This indicates that personalization consistently improves model’s performance regardless of modality. However, there is a drop in the performance of the population model when using one dataset for model training and evaluating the model using data from an unseen new user that belongs to a different dataset. Accordingly, the impact of the covariate shift is evident in this multiple-dataset scenario with the population model. These results encourage us to explore and quantify the difference in the performance of the population models and the personalized models across the different scenarios.

#### Comparison Between Models’ Performance in the Evaluation Scenarios

To assess the impact of the covariate shift on the performance of the population model and the personalized model, we first compare the performance of each individual model across the evaluation scenarios—i.e., with a single-dataset or multiple-dataset—before we then assess the performance gap between the personalized model and the population model in each scenario using Cliff’s δ effect size. In both experiments, we use Cliff’s δ effect size to quantify the difference in performance.

In Figure 5, we show Cliff’s δ effect size between the performance of the models in the different evaluation scenarios. From Figure 5, we find that the effect size for the performance of the personalized model in the two evaluation scenarios ranges between negligible and small. This means that the performance of the personalized model is consistent across the two evaluation scenarios. On the other hand, the effect size for the performance of the population model across the two evaluation scenarios ranges between medium and high for most of the modalities. These results imply that the performance of the population model changes based on the evaluation scenario.

Moreover, we compare the performance gap between the personalized model and the population model in the two evaluation scenarios. In Figure 6, we report the effect size between the performance of the two models in the single-dataset and the multiple-dataset scenario. From Figure 6a,b, we observe that the effect size between the performance of the personalized model and the population model in the multiple-dataset scenario is always larger than the effect size for the single-dataset scenario, regardless of the modality and the classifier. This implies that the difference in the performance of the population model and the personalized model is bigger in the multiple-dataset scenario than in the single-dataset scenario.

In other words, these results indicate that the performance gap between the personalized model and the population model becomes larger when different datasets are used for training and evaluation. This confirms that a mismatch in the features’ distribution between the training and evaluation sets, known as covariate shift, negatively impacts the performance of the population model on unseen users. We provide more results for the performance of the population and the personalized models across the three scenarios in the Appendix A.

## 7. Unsupervised Domain Adaptation with Population Model

In the preceding Section 6, we show that covariate shift hinders the performance of population models when using data from users in different datasets for model training and evaluation. On the other hand, we also show that personalized models are less affected by the presence of covariate shift than population models. However, researchers have shown that personalized models are difficult to implement in the real-world, since they heavily rely on user feedback to continuously adapt [13].

In this section, we propose a novel approach which bridges the gap between population and personalized models. We call this approach the *cluster-based population model* (CBPM). We use a multi-step instance-based unsupervised DA method [25] to improve the performance of the population model in the presence of covariate shift. The proposed approach leverages an unsupervised clustering algorithm with a population model, hypothesizing that narrowing the feature space for a population model by including only data from similar sleep sessions in the training set of the model can improve its performance. Moreover, the CBPM model focuses on clustering similar sleep sessions together instead of similar users, acknowledging that an individual’s behaviour can vary over time due to factors such as context, health conditions, emotional states, mental health [69], and *session-level* clustering rather than *user-level* clustering. For our novel approach, we perform experiments using the M2sleep and the BiheartS datasets in the *multiple-dataset scenario* only, since we aim to simulate real-world settings where the ML model is trained on a set of users from publicly available dataset(s) and then used by new unseen users after deployment. Figure 7 shows a demonstration of the proposed approach.

### 7.1. Configuration of the Clusters

We employ the Hierarchical Density-Based Spatial Clustering of Applications with Noise (HDBSCAN) algorithm [70] in our cluster-based population model. An ablation study is conducted to tune two key hyperparameters: minimum cluster size and minimum samples. Using the Silhouette Coefficient [71] to evaluate clustering quality, we select the configuration yielding the highest coherence based on training set feature vectors in a cross-dataset setup. Hyperparameter tuning is performed separately for each modality. Feature vectors labeled as noise by HDBSCAN are excluded. Details on the selected hyperparameters are provided in Appendix A. To implement the HDBSCAN clustering algorithm, we use the HDBSCAN Python library (https://hdbscan.readthedocs.io/en/latest/api.html) (accessed on 17 June 2025).

### 7.2. Training of Cluster-Based Population Model

After the selection of the optimal hyperparameters for each modality, using the feature vectors from the training set only, we train a population model for each cluster individually. To this end, we use both the PAC and the MLP classifiers, as in Section 6.2. In case the feature vectors in the same cluster have only a single class label, we randomly select other feature vectors with a different class label from the other clusters.

### 7.3. Evaluation of Cluster-Based Population Model

We evaluate our CBPM model using the left-out dataset. During the evaluation phase, for each feature vector in the left-out dataset, we first assign the test feature vector using the HDBSCAN clustering algorithm to one of the existing clusters. Then, we retrieve the population model that corresponds to the cluster of the feature vector. We use this population to predict the label for the test feature vector, then we evaluate the prediction using the accuracy evaluation metric, as carried out in Section 6.3. In case the HDBSCAN algorithm clusters the test feature vector as noise, we assign this test feature vector to the nearest cluster by computing the distance between the test feature vector and the centroids of all the existing clusters.

### 7.4. Experimental Results for the Cluster-Based Population Model

In this set of experiments, we compare the performance of the CBPM model with respect to the performance of the population model and the personalized model in a multiple-dataset scenario.

In Figure 8, we present the performance of the three models using the PAC and the MLP classifiers. Using Figure 8a,b, we can compare the performance of the population model, the personalized model, and the CBPM model using the PAC classifier. The results show that the performance of the CBPM model outperforms the population model using the contextual modality in both datasets. In particular, the CBPM model achieves a maximum improvement of 9.03 and 4.39 percentage points using the contextual modality for the M2sleep and the BiheartS dataset, respectively. These improvements are statistically significant compared to the population model, based on the Wilcoxon statistical test (α=0.05).

Moreover, in Figure 8c,d we compare the performance of three models using the MLP classifier. The results show that the CBPM model outperforms the population model for most of the modalities in the two datasets. In particular, we observe maximum improvement of 13.45 and 9.67 percentage points using the contextual modality in the M2sleep and the BiheartS datasets, respectively. Moreover, the CBPM model outperforms the population model in the all-sensor-features and the all-features modalities in the BiheartS dataset, with an improvement of 4.22 and 7.39 percentage points, respectively. These improvements are statistically significant when using the Wilcoxon statistical test (α=0.05). The CBPM model also outperforms the personalized model when using the contextual modality and when testing on the BiheartS dataset. Finally, we observe that the CBPM model outperforms the two baselines in most of the modalities.

These results show that performance of the CBPM model is positioned between that of the personalized and population models.

## 8. Discussion

In this section, we present the discussion and the implications of the obtained experimental results.

The first set of experiments investigate the existence of the distribution shift in personal informatics datasets collected for the same task. In this experiment, we rely on statistical analysis, similar to Meegahapola et al. [19] and Xu et al. [3], to investigate the covariate shift between the BiheartS and the M2sleep datasets. The results of the statistical analysis indicate that the majority of the features have statistically different distributions in the two datasets. Moreover, the correlation analysis shows that the associations between the features and the sleep quality labels are different in the two datasets. These results suggest the existence of covariate shift between the two datasets.

Moreover, we implement the *“Name-The-Dataset”* task, since it is suggested by several studies in the computer vision literature as an indicator of the existence of distribution shift across different domains [17,18]. The goal of the task is to analyze the capability of a classifier to distinguish between feature vectors from the two datasets. This task was also recently adopted in a human behavioural analysis study [3]. Our results show that an ML model differentiates between the feature vectors from the M2sleep dataset and the BiheartS dataset with a balanced accuracy up to 90.72%. This implies that the features in each dataset have distinguishable distributions and that each dataset has a unique identifiable signature. The results of the first set of experiments reveal the presence of a distribution shift, specifically, a covariate shift between the BiheartS and M2sleep datasets. These findings underscore that covariate shift can arise even between datasets that share similar characteristics (such as the use of wearable devices, self-reported labels, geographic region, and user demographics) and that are collected for the same task. In real-world applications, such shifts can significantly affect the performance of machine learning models trained on one dataset when deployed to new and unseen users. While the impact of covariate shift has been extensively studied in fields such as computer vision and natural language processing, it has received relatively little attention in the context of PI systems.

In the second set of experiments, we investigate the impact of the covariate shift on the performance of personalized models and population models, in a sleep quality recognition task. We extract features from the ST and the ACC modalities as in [7]. Moreover, we use features from the HR, the HRV, and the contextual modalities as in [28]. The HR and the HRV signals correlate with sleep stages and sleep quality as highlighted by several studies, e.g., [52,72,73]. We rely on self-reports to collect the ground truth for the sleep quality recognition task, similar to [7,28]. Finally, we formulate the sleep quality recognition task as a binary classification problem as in [7,28]. In addition, we use two datasets, BiHeartS and M2sleep, to estimate the impact of the covariate shift on the performance of personalized and population models in three distinct evaluation scenarios. Our experimental results indicate a decline in the performance of the population models (up to 18.54 percentage points) when trained and evaluated on data from different datasets, compared to when both sets originate from the same dataset. These findings demonstrate that the existence of covariate shift between the users in the different datasets hinders the performance of population models. In other words, training a machine learning model using data from a cohort of users with similar characteristics as the expected target user is insufficient for ML models in the context of PI systems.

On the other hand, for the personalized models, our experimental results indicate that personalized models achieve comparable performance regardless of whether the training and testing datasets originate from the same source. This suggests that personalized models effectively address both interpersonal and intra-personal variability by incorporating user-specific feedback to continuously adapt over time. These findings align with those of Gashi et al. [7], which underscore the importance of personalization in sleep quality recognition, given the subjective nature of self-reported ground truth and the inherent variability in input features across and within individuals. However, the personalized model requires feedback from unseen target users about the model’s predictions to adjust the model incrementally. This continuous user feedback is cognitively demanding and requires high engagement from the user, as highlighted by Meegahapola et al. [13].

Accordingly, in this paper, we propose a novel approach that adopts unsupervised DA to improve the performance of the population model for unseen users, without the need for the user feedback. This novel approach relies on the multi-step instance-based domain adaptation method. It integrates the HDBSCAN clustering algorithm with the population model. This cluster-based population model does not require labelled data from the target user. Instead, it relies on creating clusters out of the available training data and trains a population model for each cluster individually. Then, during the inference phase, it assigns the sleep sessions of the target user to the nearest cluster and uses the population model of the corresponding cluster to infer the sleep quality. The key novelty of the cluster-based population model (CBPM) lies in its clustering approach; instead of grouping the similar users together in the same cluster, the CBPM model uses feature vectors extracted from sleep sessions to group the same sessions together. This strategy addresses intra-personal variability by allowing sessions from the same user to be assigned to different clusters, acknowledging that an individual’s behaviour can vary over time due to factors such as context, health conditions, emotional states, and mental health [69]. These factors can influence a user’s preferences, perceptions, and physiological responses. By focusing on session-level clustering rather than user-level grouping, the CBPM model adapts more effectively to such temporal changes. Additionally, by training population models only on data from similar sleep sessions, the CBPM narrows the feature space, improving model performance. These characteristics contribute to improved performance in sleep quality recognition tasks, which inherently require personalization due to the subjectivity of self-reported ground truth and the interpersonal variability among users, as emphasized by Gashi et al. [7].

Our results show that this approach improves the performance of population models in the presence of covariate shift, with a maximum improvement of 13.45 and 9.67 percentage points using the contextual features for the M2sleep and the BiheartS datasets, respectively. Our findings highlight the potential of the unsupervised DA approach to accommodate for the interpersonal variability among users and the intra-personal variability within users. This approach eliminates the need for labelled data from the target user, instead leveraging unlabelled data. Accordingly, it reduces the burden of continuous feedback from the users that can impact the overall user experience.

## 9. Limitations and Future Work

A limitation of this study is that we focus our analysis on covariate shift. As part of our future research, we plan to investigate the existence of different types of domain shift, e.g., label shift and concept shift.

For the personalized model, we also assume that new unseen users will consistently provide feedback. The personalized model uses this feedback to adapt incrementally to the target user. In practice, however, users may not always offer consistent feedback, which could hinder the model’s ability to personalize its predictions effectively. Future work can address this by using generative artificial intelligence for data augmentation, given its potential in the image domain [74].

As part of our future research, we aim to extend this work by exploring more objective tasks, such as activity recognition, to evaluate whether the need for personalization—as emphasized in the current study—remains important in other contexts. Additionally, we plan to collect a larger and more diverse dataset, as the current BiheartS dataset is limited to only 10 users and exhibits a significant gender imbalance. With a larger dataset, we will also be able to investigate the use of deep learning classifiers, which typically require substantial data to train effectively and may offer further improvements in modelling complex patterns in wearable sensor data.

## 10. Conclusions

PI systems help users track their daily habits. These systems include ML models trained using existing datasets which are then deployed for use by unseen arbitrary users, i.e., *population models*. However, distribution differences between the training data and the data of new arbitrary users lead to a decline in these models’ performance.

In this paper, we use two datasets, the new BiHeartS and the existing M2sleep, collected from participants with comparable demographics and geographical areas and using similar wearable devices. Our experimental results confirm the existence of covariate shift between the two datasets that share similar characteristics.

Moreover, we explore the impact of covariate shift on the performance of population and personalized models. Our results highlight that personalized models effectively address covariate shift by continuously adapting to the target user, but they require ongoing user feedback, thus limiting practical applicability. On the other hand, population models are affected by covariate shift when trained and tested on different datasets, compared to when using the same dataset for training and testing.

To overcome the limitations of population models, which are affected by domain shift, and personalized models, which require ongoing user feedback, we propose the novel cluster-based population model (CBPM), which employs an unsupervised domain adaptation approach. Our results show that our CBPM approach improves population model performance by up to 13.45 percentage points in terms of accuracy when using different datasets for model training and testing, while also not requiring continuous user feedback. This method provides a practical and efficient solution, narrowing the performance gap between personalized and population models in the presence of domain shift.

## Figures and Tables

**Figure 1 sensors-25-04012-f001:**
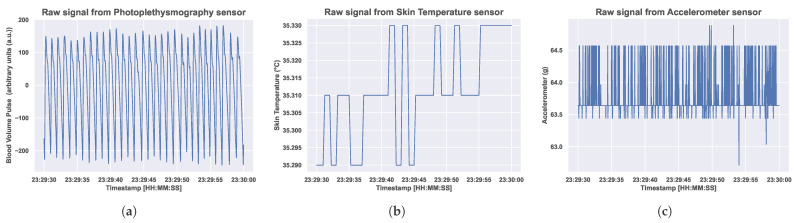
Raw signals from photoplethysmography, skin temperature, and accelerometer sensors collected using the Empatica E4 wristband wearable device. (**a**) Photoplethysmography sensor. (**b**) skin temperature sensor. (**c**) accelerometer sensor.

**Figure 2 sensors-25-04012-f002:**
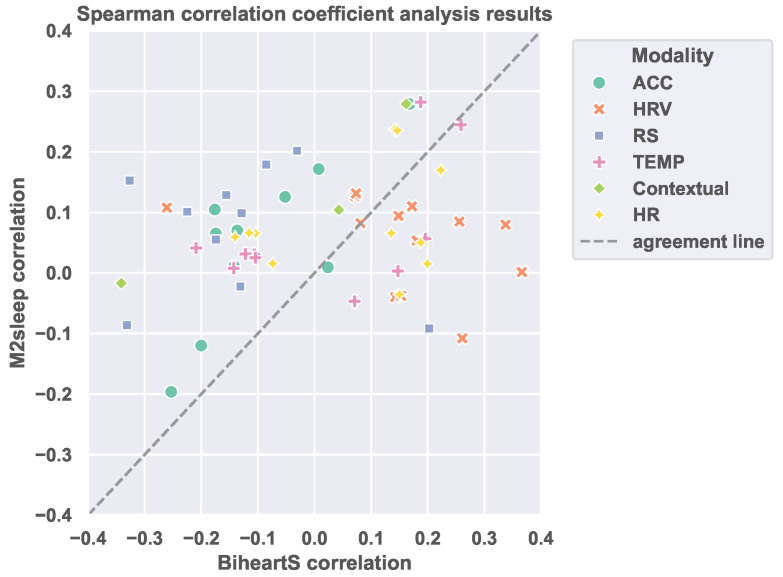
Spearman’s rank correlation coefficient between extracted features and the reported sleep quality. The diagonal line represents the agreement between correlation coefficients in both datasets.

**Figure 3 sensors-25-04012-f003:**
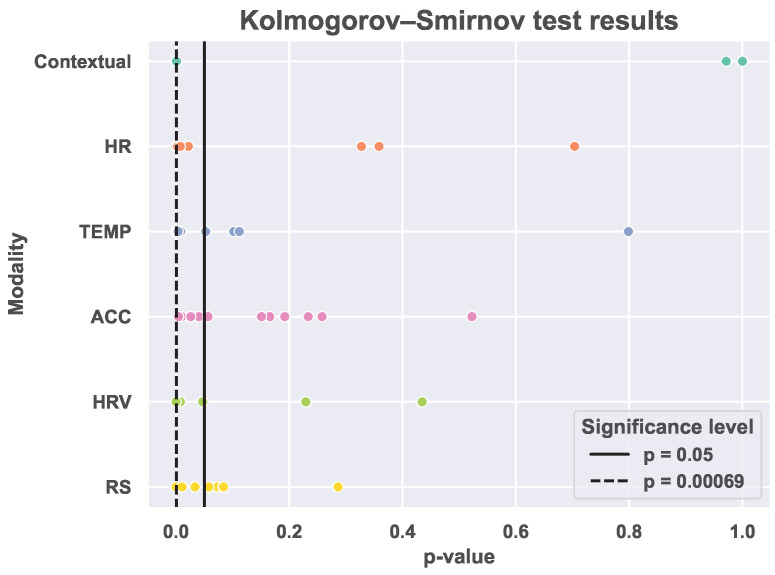
Visualization of the distribution of the *p*-value computed for each feature in the two datasets using the Kolmogorov−Smirnov non−parametric statistical test.

**Figure 4 sensors-25-04012-f004:**
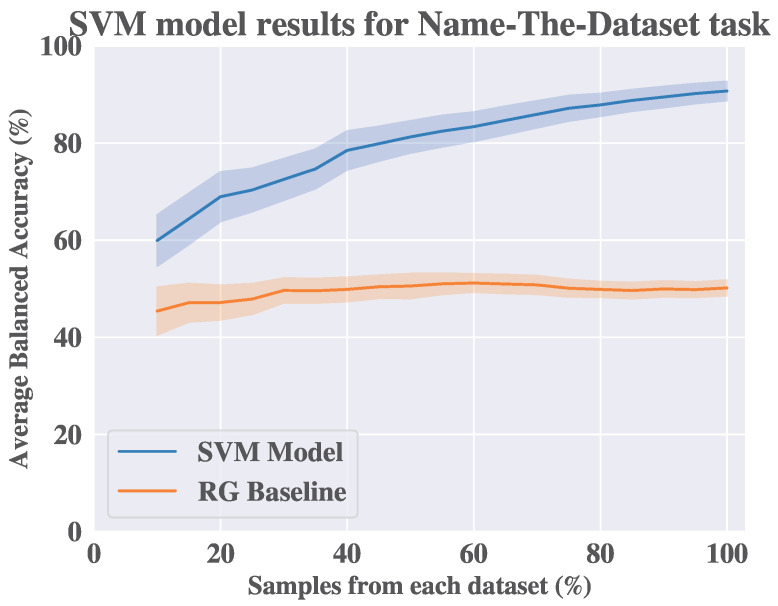
Results of the support vector machine (SVM) model and the random guess baseline in name-the-dataset ML task. The experiment is conducted multiple times with a different number of samples in each round to investigate the behaviour of the SVM model. The shadows represent the standard error for each iteration.

**Figure 5 sensors-25-04012-f005:**
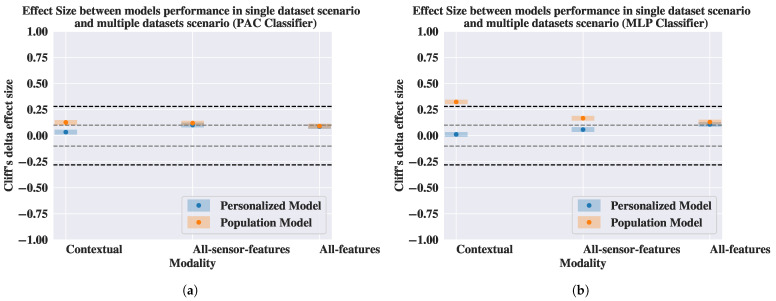
Cliff’s δ effect size between the performance of the models in two evaluation scenarios. The dashed grey and back horizontal lines represent the thresholds for the small and the medium effect size as indicated by Vargha and Delaney [68]. (**a**) Passive Aggressive (PAC) classifier. (**b**) Multilayer Perceptron (MLP) classifier.

**Figure 6 sensors-25-04012-f006:**
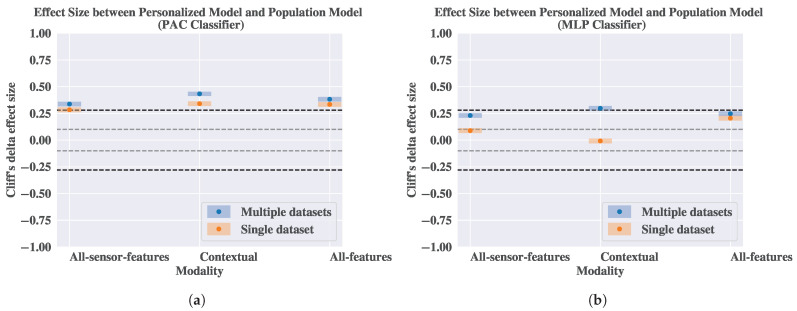
Cliff’s δ effect size between the performance of the personalized model and the population model in the single-dataset and the multiple-dataset scenarios. The dashed grey and back horizontal lines represent the thresholds for the small and the medium effect size as indicated by Vargha and Delaney [68]. (**a**) Passive Aggressive (PAC) classifier. (**b**) Multilayer Perceptron (MLP) classifier.

**Figure 7 sensors-25-04012-f007:**
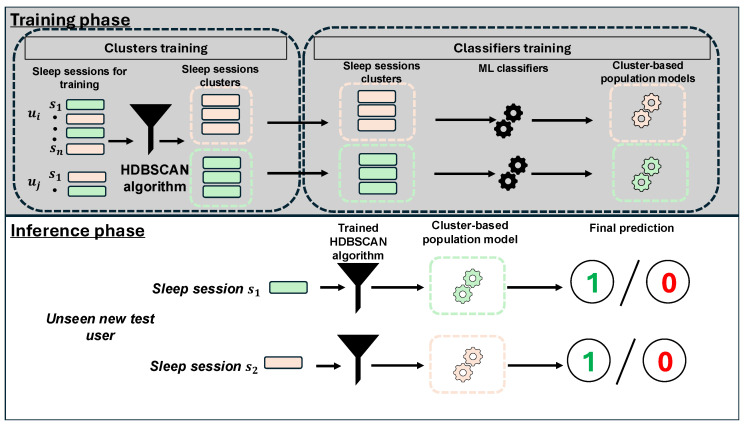
Illustration of the proposed cluster-based population model (CBPM) that integrates the Hierarchical Density-Based Spatial Clustering of Applications with Noise (HDBSCAN) clustering algorithm with the population model. In the training phase, the feature vectors of the training dataset are clustered using the HDBSCAN algorithm, and the feature vectors labeled as noise are discarded. Then, a population model is trained for each cluster independently. During the inference phase, each feature vector from the target test user is assigned to one of the existing clusters. The population model of the corresponding cluster is used to predict the binary sleep quality.

**Figure 8 sensors-25-04012-f008:**
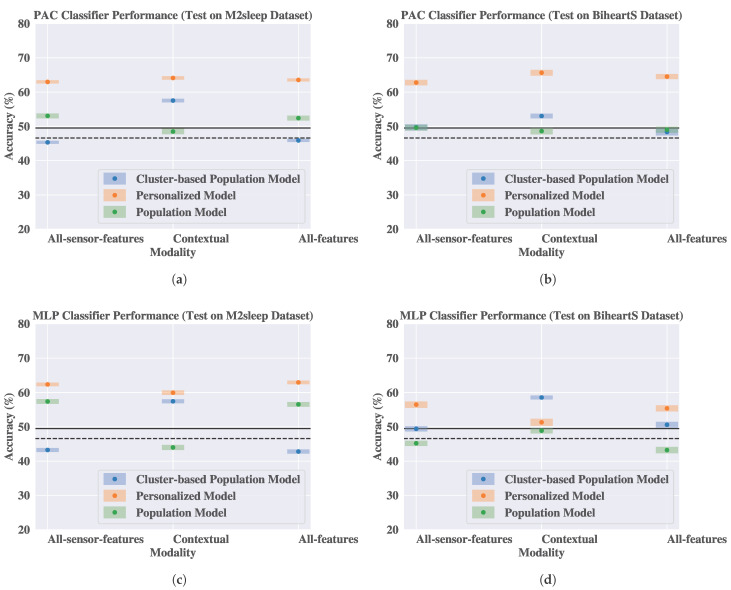
The performance of the cluster-based population model using Passive Aggressive Classifier (PAC) and Multilayer Perceptron (MLP) on the M2sleep and the BiheartS datasets. The dashed and the solid horizontal lines represent the random guess and the biased random guess baselines, respectively. (**a**) PAC classifier trained on the BiheartS dataset and tested on the M2sleep dataset. (**b**) PAC classifier trained on the M2sleep dataset and tested on the BiheartS dataset. (**c**) MLP classifier trained on the BiheartS dataset and tested on the M2sleep dataset. (**d**) MLP classifier trained on the M2sleep dataset and tested on the BiheartS dataset.

**Table 1 sensors-25-04012-t001:** Summary of the two datasets used in this paper. Symbol (✓) indicates that the dataset is collected in the wild.

Dataset	In the Wild	Users	Duration	Devices
M2sleep	✓	16	30	E4 wristband
BiheartS	✓	10	30	Two E4 wristbands and Oura ring Gen.3

**Table 2 sensors-25-04012-t002:** Summary of the seven modalities used in our experiments. The (‖) symbol represents the concatenation of two vectors.

Modality	Feature Vector (fv)
HR	$fv_{HR}\in R^{12}$
HRV	$fv_{HRV}\in R^{19}$
ST	$fv_{ST}\in R^{12}$
ACC	$fv_{ACC}\in R^{12}$
RP	$fv_{RP}\in R^{12}$
All-sensor-features	$fv_{all-sensor-features}=[fv_{HR}\|fv_{HRV}\|fv_{ST}\|fv_{ACC}\left\|fv_{RP}\right]\in R^{67}$
Contextual	$fv_{Contextual}\in R^{4}$
All-features	$fv_{all-features}=\left[fv_{all-sensor-features}\|fv_{contextual}\right]\in R^{71}$

**Table 3 sensors-25-04012-t003:** Results of the sleep quality recognition binary task for single- and multiple-dataset scenarios. We report the mean accuracy (%)±standard error. Symbols indicate statistical significance of the models: (*) vs. random guess, (**) vs. random and biased random guess, (†) vs. biased random guess. Bold values show significant differences between population and personalized models within the same modality and scenario. All statistical comparisons are carried out using the Wilcoxon paired non-parametric test (α=0.05).

Passive Aggressive Classifier Performance				
**Modality**	**Single-dataset Scenario**		**Multiple-dataset Scenario**	
	**Population Model**	**Personalized Model**	**Population Model**	**Personalized Model**
**Test on the M2sleep dataset**				
All-sensor-features	51.25 ± 0.01 *	**63.40 ± 0.00** **	53.06 ± 0.01 **	**62.97 ± 0.00** **
Contextual	56.57 ± 0.01 **	**64.48 ± 0.01** **	48.50 ± 0.01	**64.12 ± 0.00** **
All-features	50.59 ± 0.01 *	**63.75 ± 0.00** **	52.41 ± 0.01 **	**63.55 ± 0.00** **
BRG baseline	53.48 ± 0.01	51.67 ± 0.00	46.93 ± 0.01	49.24 ± 0.00
RG baseline	47.44 ± 0.01			
**Test on the BiheartS dataset**				
All-sensor-features	64.41 ± 0.01 **	**69.85 ± 0.01** **	49.63 ± 0.01 †	**62.77 ± 0.01** **
Contextual	48.27 ± 0.01	**67.12 ± 0.01** **	48.62 ± 0.01	**65.66 ± 0.01** **
All-features	62.02 ± 0.01 **	**70.39 ± 0.01** **	49.02 ± 0.01 †	**64.52 ± 0.01** **
BRG baseline	51.60 ± 0.01	51.12 ± 0.01	46.06 ± 0.01	47.51 ± 0.01
RG baseline	52.81 ± 0.01			
**Multilayer Perceptron Classifier Performance**				
**Test on the M2sleep dataset**				
**Modality**	**Single-dataset Scenario**		**Multiple-dataset Scenario**	
	Population Model	Personalized Model	Population Model	Personalized Model
All-sensor-features	55.53 ± 0.01 *	**63.74 ± 0.01** **	57.40 ± 0.01 **	**62.37 ± 0.01** **
Contextual	60.69 ± 0.01 **	60.42 ± 0.01 **	43.99 ± 0.01	**59.96 ± 0.01** **
All-features	54.83 ± 0.01 *	**64.95 ± 0.01** **	56.55 ± 0.01 **	**62.97 ± 0.01** **
BRG Baseline	53.48 ± 0.01	51.67 ± 0.00	46.93 ± 0.01	49.24 ± 0.0
RG Baseline	47.44 ± 0.01			
**Test on the BiheartS dataset**				
All-sensor-features	**63.73 ± 0.01** **	59.03 ± 0.01†	45.19 ± 0.01	**56.48 ± 0.01** †
Contextual	51.17 ± 0.01	52.26 ± 0.01	48.89 ± 0.01†	51.32 ± 0.01†
All-features	59.85 ± 0.01 **	**61.28 ± 0.01** **	43.21 ± 0.01	**55.39 ± 0.01** †
BRG Baseline	51.60 ± 0.01	51.12 ± 0.01	46.08 ± 0.01	47.51 ± 0.01
RG Baseline	52.81 ± 0.01			

## Data Availability

Both datasets used in this work are available for the research community upon signing a data sharing agreement request. In particular, for the BiheartS dataset, please contact the corresponding author. Also, for the M2sleep dataset, more details can be found in [7].

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
