# Peer review of "The Impact of Domain Shift on Predicting Perceived Sleep Quality from Wearables"

_sensors, 2025, doi:10.3390/s25134012_

Round 1
Reviewer 1 Report
Comments and Suggestions for Authors
(1) The title, "Personalization is All You Need," appears overly ambitious. From a causal inference perspective, it’s just an excuse of not analyzing covariables.
(2) The writing is comprehensive, but too much. The authors should focus on clarity—conciseness enhances readability and impact.
(3) Previous reported accuracy for sleep stage classification is much higher (e.g., 80% in prior studies [1]), which contrasts sharply with the low performance of the proposed "single quality index." This discrepancy warrants explanation. Furthermore, if subjective self-reports serve as the reference standard, the utility of wearable monitoring becomes questionable.
(4) To mitigate confounding, personalization analyses should disentangle biometric markers from objective measurements (such as weight).
(5) Why were respiration patterns not included as features? This question ties back to the fist comment. If apnea was not considered, how can you claim 'xx is all you need'?
- Sridhar, N., et al., Deep learning for automated sleep staging using instantaneous heart rate. npj Digital Medicine, 2020. 3(1): p. 106.
Reviewer 2 Report
Comments and Suggestions for Authors
The article titled “Personalization is All You Need: The Impact of Domain Shift on the Performance of Sleep Quality Recognition Models” develops a novel Cluster-Based population Model 11 (CBPM,) which achieves accuracy improvements. The study is interesting and exhibits some direct social impact. However, some detailed explanation needs to be included for further improvement. Additionally, a major concern is whether the topic fits well within the core scope of the Sensors journal, as the primary focus is on sensor technologies rather than machine learning methodologies.
Major points:
- While PAC and MLP are common classifiers, the rationale behind choosing only these two models is not sufficiently explained. The authors are encouraged to elaborate on why these models were selected and to consider including additional one or two representative models to strengthen the validation of CBPM’s performance.
- Consider adding a figure illustrating the raw sleep data collected from wearable devices to enhance methodological transparency.
- Consider including a schematic illustration of CBPM mode’s analysis process and working principles.
- Although the CBPM model demonstrates improved accuracy, the underlying mechanisms contributing to this improvement are not fully explored. A more in-depth explanation or theoretical discussion of why CBPM yields better performance would enhance the scientific value of the work.
- The BiheartS dataset includes nine male and only one female participant, which introduces a significant gender imbalance. The potential impact of gender differences on physiological signals and sleep quality should be discussed
- The sleep quality label is derived from self-report, which is subjective and may cause label noise. A discussion of the reliability of these labels and possible mitigation strategies should be included.
- Please ensure consistent font sizes across all figures, particularly for axis labels. In Figures 3 and 5, the current font size is too small and affects readability.
Round 2
Reviewer 2 Report
Comments and Suggestions for Authors
The author has addressed all my concerns in detail, and the revised manuscript is suitable for publication.
Author Response
We sincerely thank the reviewer for their positive feedback and for acknowledging our revisions. We appreciate your careful evaluation and are pleased that the revised manuscript has addressed your concerns. Thank you for your recommendation and support.